# How to Use Nebulized Antibiotics in Severe Respiratory Infections

**DOI:** 10.3390/antibiotics12020267

**Published:** 2023-01-28

**Authors:** Julie Gorham, Fabio S. Taccone, Maya Hites

**Affiliations:** 1Department of Intensive Care Unit, Hôpital Universitaire de Bruxelles (HUB), 1070 Brussels, Belgium; 2Clinic of Infectious Diseases, Hôpital Universitaire de Bruxelles (HUB), 1070 Brussels, Belgium

**Keywords:** nebulized antibiotic, respiratory tract infection, pharmacokinetic/pharmacodynamic

## Abstract

Difficult-to-treat pulmonary infections caused by multidrug-resistant (MDR) pathogens are of great concern because their incidence continues to increase worldwide and they are associated with high morbidity and mortality. Nebulized antibiotics are increasingly being used in this context. The advantages of the administration of a nebulized antibiotic in respiratory tract infections due to MDR include the potential to deliver higher drug concentrations to the site of infection, thus minimizing the systemic adverse effects observed with the use of parenteral or oral antibiotic agents. However, there is an inconsistency between the large amount of experimental evidence supporting the administration of nebulized antibiotics and the paucity of clinical studies confirming the efficacy and safety of these drugs. In this narrative review, we describe the current evidence on the use of nebulized antibiotics for the treatment of severe respiratory infections.

## 1. Introduction

Nebulized antibiotics have been used to treat respiratory tract infections over the last 70 years but are not considered the first choice in this setting. Nevertheless, there has been a recent resurgence in interest for this administration route [1] because of the emergence of multidrug-resistant (MDR) strains as causative pathogens of severe respiratory tract infections. MDR bacteria are bacteria resistant to at least one antibiotic molecule in three or more antimicrobial categories. Biofilm production, enzymatic inactivation of the antibiotic, active efflux pumps and modification of the target of the antibiotic are different antibiotic resistance mechanisms developed by bacteria.

Nebulized antibiotics are mainly used in critically ill patients to treat ventilator-associated pneumonia (VAP), which is a serious infection that develops in approximately one-third of patients who receive mechanical ventilation for more than 48 h and is responsible for over 50% of antibiotic prescriptions in the intensive care unit (ICU) [2]. VAP is associated with high costs [3,4] and considerable morbidity and mortality [5,6], particularly when caused by MDR Gram-negative bacteria (GNB) [7]. VAP is also associated with increased durations of mechanical ventilation and lengths of ICU and hospital stay. The pathogens responsible for VAP are primarily GNB; however, *Staphylococcus aureus* is also sometimes involved. The most prescribed nebulized antibiotics are colistin and aminoglycosides. Many theoretical advantages of nebulized antibiotics have been proposed, such as higher concentrations at the site of infection and less systemic exposure [1]. These potential benefits could help reduce the emergence of antibiotic resistance and minimize adverse effects [8].

In this narrative review, we review the use of nebulized antibiotics for severe respiratory tract infections in the ICU setting. We will discuss the general pharmacokinetic/pharmacodynamic (PK/PD) principles of treating infections, how nebulized antibiotics are delivered, and the clinical evidence available that supports the use of nebulized antibiotics. We will also provide some recommendations on how to use nebulized antibiotics in this setting.

### 1.1. Pharmacodynamic Targets to Treat Severe Infections

To eradicate a pathogen, concentrations of the antibiotic at the site of infection need to exceed the pathogen’s minimum inhibitory concentration (MIC). The MIC represents the lowest antibiotic concentration that prevents visible bacterial growth with a standardized inoculum in vitro and serves as the basis for assessing whether a pathogen is susceptible or resistant to a given antibiotic. Epithelial lining fluid (ELF) antibiotic concentration, and more particularly the “pulmonary penetration ratio”, defined as the ratio between drug exposure in the ELF and plasma, is an important determinant of the efficacy of the treatment of bacterial pneumonia [9].

A few studies [10,11,12,13,14,15] have investigated the PD targets of antibiotics in the ELF after intravenous administration in critically ill patients. These reports have shown that several drugs have poor penetration into the ELF when administered intravenously, due to limited alveolo-capillary barrier permeability and host- and drug-related factors [16]. As such, standard antibiotic regimens are unlikely to achieve the optimal C_max_/MIC or AUC/MIC ratios associated with an increased probability of favorable clinical response [17]. As an example, one study showed that tobramycin had approximately 12% penetration into the ELF, compared to the serum C_max_ [11]; in another study, colistin could not be detected in the ELF of critically ill patients with VAP after 2 days of intravenous drug administration [10]. Inadequate ELF concentrations may increase the risk of therapeutic failure, in addition to causing systemic toxicity and/or the emergence of resistance [18]. 

The risk of not achieving adequate antibiotic concentrations in the ELF is further increased when trying to treat drug resistant GNB. In particular, MDR or extensive drug resistance (XDR, defined as the non-susceptibility of one bacterial species to all antimicrobial agents except for two or fewer antimicrobial categories). GNB infections, mostly *Pseudomonas aeruginosa* and *Acinetobacter baumannii*, are currently the most common in causing VAP in Europe and Asia [18,19]. In a prospective study conducted in 10 Asian countries from 2008 to 2009, the MDR rates of these strains were 82% and 43%, respectively, and the XDR rates were 51% and 5%, respectively [20]. Colistin and amikacin often remain the only antimicrobial agents available against these pathogens [20,21]. In particular, the nebulized administration of these drugs has been advocated to achieve high local concentrations of antibiotics, well above the MIC of the causative pathogens. Unlike the systemic administration of antibiotics, aerosolized antibiotics avoid the need for drug diffusion across the blood–alveolar barrier and can directly reach the infected alveolar spaces. Numerous studies [22,23,24,25,26,27,28], particularly in animals, have quantified the concentration of antibiotics in the lungs after the administration of nebulized antibiotics. In ventilated piglets with *Escherichia coli* VAP, Goldstein et al. [2] showed that amikacin concentrations in the lungs were much higher than the MIC of the pathogen with using nebulized compared to intravenous administration. In another animal study [28], the administration of nebulized vancomycin resulted in higher lung tissue concentrations than administration via the intravenous route in mechanically ventilated healthy piglets. In addition, relatively high antibiotic concentrations have also been observed in non-aerated lung regions, probably due to diffusion through the bronchiolar mucosa to closer consolidated alveoli.

>Human studies have reported the same results: nebulized colistin achieved drug concentrations that exceeded *P. aeruginosa*’s European Committee on Antimicrobial Susceptibility Testing (EUCAST) MIC breakpoint [29]. In one study, amikacin ELF concentrations following a nebulized dose of 400 mg were approximately 28- to 35-fold higher than concentrations obtained by intravenous administration; Luyt et al. [26] measured very high concentrations of amikacin in the ELF after inhaled administration, systematically exceeding the usual MIC of pathogens in patients with GNB VAP. Furthermore, serum amikacin concentrations were still within the non-toxic therapeutic ranges.

However, the PKs of nebulized antibiotics are not completely understood, mainly because of the difficulty in assessing lung interstitial antibiotic concentrations in human studies. One of the possible procedures to assess the concentration of antibiotics at the most distal pulmonary level is the assessment of the ELF by bronchoalveolar lavage (BAL). However, there are several limitations to this technique, including contamination of the bronchoscope by bronchial secretions during the bronchoalveolar procedure and dilution due to the instillation of fluid into the ELF resulting in a skewed interpretation of ELF concentrations following antibiotic nebulization [28,29,30]. Other approaches include sputum analysis, lung microdialysis, and lung tissue biopsies [31,32], which remain difficult or are very invasive procedures. 

### 1.2. Nebulized Antibiotics: Technical Issues

Nebulizers are used to convert liquid into small droplets that can be inhaled into the lower respiratory tract. The aerosol is transported by the gas flow inhaled by the patient. Due to the branching of the airways, the total cross-sectional area increases with each bronchial bifurcation and consequently the speed of the gas flow decreases from bifurcation to bifurcation (i.e., from the trachea to the terminal bronchioles). Three mechanisms are involved in the deposition of inhaled antibiotics: impaction, sedimentation, and diffusion. Impaction of the droplet occurs when a bifurcation stops the continuation of its course. This phenomenon is linked to the inertia of the particles and therefore primarily concerns particles of high mass and high velocity. Thus, the largest particles will tend to settle by impaction in the proximal airways or in the mechanical ventilation circuit. Sedimentation is related to the deposition of particles in the airways under the influence of gravity; this phenomenon again concerns heavy particles. Diffusion leads to the deposition of the smallest particles at the level of the most distal airways [33]. The smallest droplets are partly exhaled without being deposited in the airways. As such, several factors influence the pulmonary penetration of nebulized antibiotics, such as the type of nebulizer used and the size of the particles (i.e., particles with a diameter greater than 5 μm are mainly deposited in the ventilation circuit and/or the upper airways; particles of 3–5 μm are deposited in the proximal bronchi; and particles of 1–3 μm are deposited in the alveoli and terminal bronchioles) [34]. 

Not all types of nebulizer deliver aerosol particles with the same efficiency, so the type of aerosol chosen is important. There are three nebulization techniques: jet, vibrating mesh, and ultrasonic nebulizers. Vibrating mesh and ultrasonic nebulizers should be preferred to jet nebulizers, which produce a highly turbulent flow, permitting fewer particles to be deposited in the lung parenchyma [35]. However, ultrasonic nebulizers have the disadvantage of increasing the temperature of the antibiotic solution, potentially resulting in the chemical alteration of the molecule used, and are more expensive. Recent studies tend to favor vibrating systems, with which the liquid solution is placed above a membrane that vibrates in response to an electrical impulse [33,34]. With this type of nebulizer, retention in the nebulizer is negligible and there is no increase in temperature. The respirator humidifier filter should be removed during nebulization and the nebulizer placed on the inspiratory circuit 15 to 40 cm from the Y-piece [36]. Ventilator and circuit connections should not have obtuse angles, which can impair aerosol drug delivery. The ventilatory parameters of the respirator also play an important role. Spontaneous ventilator modes are associated with high turbulent inspiratory flow and consequently reduce drug delivery to the lung. It is therefore preferable to ventilate the patient in a controlled volume with a low constant inspiratory flow (e.g., ventilate the patient with a tidal volume greater than 500 mL and a high inspiratory time) and avoid asynchronies [37]. Sedation may be necessary in order to avoid ventilator–patient asynchrony.

### 1.3. Existing Evidence on the Efficacy of Inhaled Antibiotics

#### 1.3.1. Ceftazidime

In an animal study comparing the effects of the nebulized and intravenous administration of ceftazidime on lung tissue deposition and antibacterial efficacy in ventilated piglets with pneumonia caused by *P. aeruginosa* with reduced susceptibility to ceftazidime, nebulized ceftazidime provided more effective bacterial killing [38]. In a prospective randomized study of 40 patients with VAP caused by *P. aeruginosa*, Lu et al. [39] compared the efficacy of dual therapy with intravenous ceftazidime and amikacin with the efficacy of an exclusively nebulized therapy. After 8 days of antibiotic administration, the cure rate, duration of mechanical ventilation, ICU length of stay, and mortality were not statistically different between the two arms. The incidence of VAP recurrence was also similar in the two groups. However, in the nebulized group, *Pseudomonas* regrowth or persistence was caused exclusively by susceptible strains, whereas in the intravenous group half of the strains had become intermediate or resistant to one or both drugs. 

#### 1.3.2. Fosfomycin

In a prospective, randomized study in pigs [40] with severe pneumonia due to *P. aeruginosa* resistant to amikacin and fosfomycin but susceptible to meropenem, intravenous meropenem was compared to different combinations of nebulized antibiotics. The pigs were randomized to receive either nebulized saline solution four times a day (QID), nebulized amikacin QID, nebulized fosfomycin QID, intravenous meropenem three times a day (TID), nebulized amikacin and fosfomycin QID, or nebulized amikacin and fosfomycin QID with intravenous meropenem TID. This study demonstrated that the efficacy of the nebulized antibiotics was greatest in tracheal secretions but that intravenous meropenem was needed to reduce the bacterial loads of *P. aeruginosa* in lung tissue. 

#### 1.3.3. Amikacin and Tobramycin

Several studies [41,42,43,44,45,46,47,48,49] have assessed the efficacy and safety of nebulized aminoglycosides and report conflicting results. A retrospective observational study [41] in 22 ventilated surgical ICU patients with GNB VAP who received nebulized aminoglycosides (either tobramycin or amikacin) as an adjunct to systemic therapy showed a clinical resolution of pneumonia and more rapid weaning from mechanical ventilation. In a retrospective single-center cohort study of 93 patients with *P. aeruginosa* and *A. baumannii* VAP, the clinical outcomes in patients treated with intravenous antibiotics and adjunctive nebulized antibiotics (150 mg colistin or 300 mg tobramycin inhaled twice daily) were compared [46]. Thirty-day mortality was significantly lower in the group of patients that received a nebulized antibiotic. In a retrospective study [46] of patients receiving adjunctive nebulized antibiotics (mostly tobramycin) for VAP caused by *P. aeruginosa* and/or *A. baumannii*, clinical and microbiological success were achieved in approximately 70% of patients. Notably, in the 20 episodes of VAP in which treatment with systemic antibiotics failed, clinical success was subsequently achieved in 85% of cases after the addition of nebulized antibiotics. In another study including cancer patients with GNB-related VAP [47], 16 patients who received nebulized aminoglycosides or colistin were compared to 16 patients who received the same agents intravenously. All the patients in the group treated with nebulized antibiotics had complete clinical resolution, compared to 55% in the intravenous group. In a randomized, double-blind pilot study in 10 patients, the efficacy and safety of nebulized tobramycin as an adjunct to systemic treatment [48] in the treatment of susceptible *P. aeruginosa* or *Acinetobacter* spp. VAP was evaluated. All patients in the group who received nebulized tobramycin showed a clinical response at 28 days, whereas only 60% in the other group responded to therapy. In a prospective, randomized controlled single-center study [44], 133 post-cardiac surgery patients with nosocomial pneumonia caused by MDR GNB, were allocated to intravenous amikacin (20 mg/kg once daily) or nebulized amikacin (400 mg twice daily); in both groups, intravenous piperacillin/tazobactam was administered empirically. The nebulized group had significantly shorter ICU stays, a shorter time to reach complete clinical cure, fewer ventilator days, fewer days on amikacin treatment, and less nephrotoxicity than the intravenous group, but there was no difference in mortality. In three RCTs (Table 1) on the use of nebulized aminoglycosides combined with intravenous antibiotics to treat VAP [42,43,49]—the IASIS, INHALE, and VAPORISE trials—nebulized aminoglycosides showed no survival benefit compared to standard therapy. The VAPORISE trial [43] was a prospective double-blind RCT on VAP, performed in a single center where patients were randomized to receive nebulized tobramycin and standard intravenous antibiotic therapy for 8 days or to a control group that received placebo nebulization and standard intravenous antibiotic treatment for 8 days. The study was terminated prematurely due to insufficient patient inclusion. In the 26 patients included, there was no difference in 30-day mortality (31% in both groups). In the IASIS RCT [49], 143 patients were randomized to receive, in addition to standard-of-care antibiotics, nebulized amikacin (300 mg) plus fosfomycin (120 mg) or placebo for the treatment of GNB VAP. Adjunctive aerosol therapy was ineffective in improving clinical outcomes (mortality 24% vs. 17%; *p* = 0.32), despite reducing bacterial burden. However, patients were only enrolled to receive nebulized amikacin and fosfomycin within 72 h of the initiation of intravenous meropenem, which may have resulted in a potential survivor bias and confounded the potential efficacy of the nebulized antibiotic combination. In the international multicenter INHALE study [42], more than 800 mechanically ventilated patients with GNB VAP were randomized to receive nebulized amikacin or placebo in addition to standard-of-care intravenous antibiotics. There was no benefit of the nebulized antibiotics on survival, early clinical cure rate, days on mechanical ventilation, or days in the ICU. However, there are potential biases in these studies, including heterogeneous populations, the origin of the pneumonia, the infecting pathogen, the nature of intravenous standard-of-care therapy, the type of nebulizer delivery system used, and the ventilator settings. The IASIS, INHALE, and VAPORISE trials may have failed to show a clinical benefit of combining nebulized aminoglycosides with intravenous beta-lactams for treating VAP due to the inclusion of many patients with VAP due to susceptible pathogens, the administration of probably sub-optimal low nebulized aminoglycoside doses, and the non-optimal ventilator settings used. 

#### 1.3.4. Colistin

Several studies [14,20,30,50,51,52,53,54,55,56,57,58,59,60,61,62,63,64,65,66,67,68,69,70,71,72] have evaluated the use of nebulized colistin as a treatment for MDR VAP and ventilator-associated tracheobronchitis (VAT), including randomized controlled trials (RCTs) [66,70,71]. Some studies evaluated the administration of nebulized colistin alone and others in combination with intravenous administration. The daily dose of nebulized colistin and treatment duration also varied across studies. In one RCT, 100 patients with MDR VAP were randomized to receive systemic antibiotics in combination with either nebulized saline solution or nebulized colistin [70]. Patients in the colistin group had significantly better microbiological outcomes compared to patients in the control group (60.9% versus 38.2%, *p* = 0.03), but there was no statistically significant difference in a favorable clinical outcome (51% vs. 53%, *p* = 0.82). In another RCT [66], the efficacy of colistin, administered in nebulized or intravenous form, alone or in addition to intravenous beta-lactam antibiotic therapy, was compared in patients with VAP caused by MDR GNB. When administered as monotherapy or in combination, patients who received nebulized colistin had no statistically significant benefit in terms of clinical efficacy compared to patients who did not receive nebulized colistin, but patients in the nebulized colistin group had a significantly larger improvement in the PaO_2_/FiO_2_ ratio (349 vs. 316 at day 14, *p* = 0.012), a shortened time to bacterial eradication (9.89 vs. 11.26 days, *p* = 0.023), and earlier weaning from mechanical ventilation. A meta-analysis [73] of 12 studies reported the effectiveness of nebulized colistin as monotherapy for respiratory tract infections due to MDR or colistin-only susceptible GNB, with a clinical and microbiological success rate of 70%. Two meta-analyses showed better clinical and microbiological responses and lower infection-related mortality in patients receiving the association of intravenous and nebulized colistin as treatment for VAP and VAT caused by MDR GNB [74,75] compared to patients receiving intravenous therapy alone. However, another meta-analysis [76], in which the combination of intravenous and nebulized treatment was compared to intravenous monotherapy in adult patients with lower tract infections due to MDR GNB did not confirm these benefits. Some of the limitations of these meta-analyses are the retrospective nature of many of the included studies, their heterogeneous protocols, the lack of optimization of the technique of nebulization, and the variability of dosing. Furthermore, the nebulization of colistin may be an efficient treatment for MDR GNB VAP and VAT but further studies are required to determine whether nebulized colistin is equivalent or superior to intravenous treatment. 

#### 1.3.5. Vancomycin 

In a non-comparative study [77], 21 critically ill ventilated patients with methicillin-resistant *S. aureus* (MRSA) VAP were treated with a 7-day course of endotracheal vancomycin, intravenous linezolid plus rifampicin, nasal mupirocin, and oropharyngeal and cutaneous decontamination. A clinical cure rate of more than 95% at the end of the treatment was reported and the treatment was also effective for MRSA eradication. In an RCT of critically ill intubated patients with VAT [78], the therapeutic effect of nebulized vancomycin and/or gentamicin was evaluated. Nebulized antibiotics significantly decreased the development of VAP and other signs and symptoms of respiratory infection, facilitated weaning, and reduced bacterial resistance and the use of systemic antibiotics for new or persistent infections compared to the placebo, but the mortality rate was not significantly different. In another RCT [8], the use of nebulized vancomycin and/or nebulized aminoglycoside or placebo for 14 days in addition to a systemic treatment were compared in mechanically ventilated patients with VAP who were at high risk for MDR pathogens. Patients receiving nebulized therapy were more likely to have the pathogen eradicated from sputum cultures (96% vs. 9%; *p* < 0.001) and also showed significant clinical improvement at the end of treatment than those in the placebo group. No significant difference was demonstrated regarding mortality or the duration of mechanical ventilation. In a prospective non-comparative study [79] of 20 mechanically ventilated patients (>48 h) receiving intravenous vancomycin for MRSA pneumonia and nebulized vancomycin (250 mg every 12 h for 5 days), 65% of the patients showed clinical cure or improvement. Microbiological eradication of MRSA was confirmed in 70% of cases, which was greater than in results obtained from the systemic administration of vancomycin [80]. 

### 1.4. Adverse Effects of Nebulized Antibiotics

It is assumed that the nebulized administration of antibiotics can limit the nephrotoxicity of drugs such as aminoglycosides or colistin because of the lower systemic passage. Indeed, in some studies [66,76], a lower incidence of renal failure was observed in the group receiving nebulized antibiotics (colistin or amikacin) compared to the intravenous route. This potential benefit has not been, however, observed in all studies [81], especially in studies evaluating nebulized colistin in addition to intravenous colistin, because of the potential nephrotoxicity of systemically administered antibiotics. The inhalation route does not protect patients from impaired renal function due to the systemic passage of the drug; however, systemic diffusion is predominantly observed with aminoglycosides and does not appear to aggravate renal function. Indeed, in a systematic review [74] conducted in critically ill adults receiving invasive mechanical ventilation, no increased risk of nephrotoxicity was reported when adding nebulized antibiotics to intravenous therapy compared to patients receiving only intravenous antibiotics. However, in patients with acute renal failure [82], some cases of renal toxicity have been described after the administration of aminoglycoside aerosols. Nevertheless, nebulized doses of amikacin up to 60 mg/kg have been well tolerated in patients with chronic kidney disease [83]. 

Cough, bronchospasm, wheezing, desaturation, and hypoxemia are among the most frequently reported pulmonary adverse effects during the administration of nebulized antibiotics [84,85,86], especially when they are administered to patients with severe hypoxemia [86]. Colistin is more often implicated in these effects than other antibiotics, and some formulations of colistin are more toxic than others [87]. Complications seem less frequent when the patient is receiving mechanical ventilation, compared to spontaneous ventilation. Obstruction of the ventilation circuit and of the intubation tube can also occur in the event of prolonged nebulization. Nebulization can cause irritation and an inflammatory reaction of the airways, which are sources of bronchospasm. In addition, sedation may be necessary to avoid ventilator–patient asynchrony, which could potentially increase the duration of mechanical ventilation.

## 2. Conclusions

The 2016 Infectious Diseases Society of America and the American Thoracic Society guidelines on the management for hospital-acquired pneumonia and VAP recommend adding nebulized antibiotics to systemic treatment in patients with VAP due to MDR GNB that are susceptible to only aminoglycosides or polymyxins (weak recommendation, low-quality evidence). This treatment may be considered as a last-resort treatment in patients who are not responding to intravenous therapy alone, regardless of the susceptibility of the infecting pathogen organism being MDR [88]. By contrast, in 2017, the European Society of Clinical Microbiology and Infectious Diseases recommended avoiding the use of nebulized antibiotics for the treatment of respiratory infections in adults receiving invasive mechanical ventilation due to the risk of toxicity and the lack of strong evidence of their efficacy [86]. In clinical practice, indications should be discussed on a case-by-case basis. For example, in a frail patient who develops VAP with a MDR bacteria, who is possibly immunocompromised, and has a high risk of therapeutic failure, nebulized antibiotic therapy could be considered in order to increase the possibility of achieving very high drug concentrations at the site of the infection (Figure 1) and to facilitate therapeutic success, especially after first-line treatment failure. It should be noted that in this situation it seems logical to systematically associate maximum intravenous treatment. In other patients with VAP due to a susceptible pathogen, it is difficult to identify a clear clinical benefit from the nebulized administration of antibiotics, given the simplicity and known effectiveness of intravenous antibiotic therapy. 

## Figures and Tables

**Figure 1 antibiotics-12-00267-f001:**
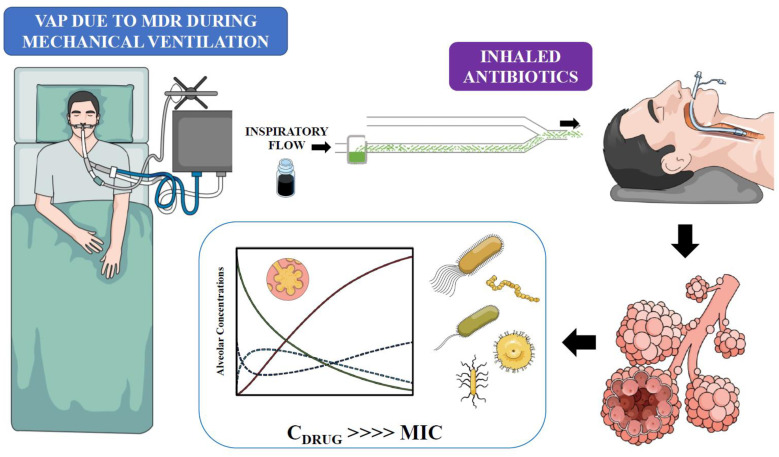
A simplified approach explaining the use of inhaled antibiotics for ventilator-associated pneumonia (VAP) due to multidrug-resistant (MDR) pathogens. Inhaled antibiotics will provide very high drug concentrations (C_DRUG_) in the alveolar space, which by far exceed the minimal inhibitory concentration (MIC) of the pathogen.

**Table 1 antibiotics-12-00267-t001:** Key features of three randomized controlled trials on the use of nebulized aminoglycosides combined with intravenous antibiotics to treat ventilator-associated pneumonia (VAP).

	IASIIASIS [49]	INHALE [42]	VAPORISE [43]
Type of study	Randomized, double-blind, placebo-controlled, phase 2, multicenter study	Randomized, double-blind, placebo-controlled, phase 3, multicenter study	Randomized, double-blind, placebo-controlled, single-center study; terminated prematurely due to insufficient inclusion
Number of patients	143	712	26
VAP pathogens	MDR GNB (29% placebo group–45% treatment group)	MDR GNB (44% placebo group–49% treatment group)	MDR GNB (0%)
Placebo group	IV-lactam + nebulized saline	IV-lactam + IV fluoroquinolone or amikacin + nebulized saline	IV-lactam + IV fluoroquinolone + nebulized saline
Treatment group	IV-lactam + nebulized amikacin/fosfomycin	IV-lactam + IV fluoroquinolone or amikacin + nebulized amikacin	IV-lactam + nebulized tobramycin
Doses of nebulized antibiotics	Amikacin 300 mg + fosfomycin 120 mg every 12 h for 10 days	Amikacin 400 mg every 12 h for 10 days	Tobramycin 300 mg every 12 h for 8 days
Type of nebulizer	Non-synchronized mesh nebulizer	Synchronized inhalation mesh nebulizer	Non-synchronized mesh nebulizer
Ventilator setting optimization	No	No	No
Primary aim	Change in CPIS after 10 days of treatment	Mortality at Days 28–32	Treatment failure at Day 4
Results	No difference between groups (*p* = 0.70)	77% placebo group vs. 75% treatment group	31% treatment group vs. 62% placebo group
Mortality rate	At Day 28: 24% treatment group vs. 17% placebo group (*p* = 0.32)	No difference in survival (77% placebo group vs. 75% treatment group)	At Day 30: no difference (31% in both groups)
Adverse events	Comparable in both groups	Comparable in both groups (84% placebo group vs. 84% treatment group)	NR

CPIS—clinical pulmonary infection score; NR—not reported; MDR—multidrug resistance; GNB—Gram-negative bacteria; IV—intravenous.

## Data Availability

Not applicable.

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
