# Peer review of "How to Use Nebulized Antibiotics in Severe Respiratory Infections"

_antibiotics, 2023, doi:10.3390/antibiotics12020267_

Round 1

Reviewer 1 Report

The review discussed the advantages of administration of a nebulized antibiotic in respiratory tract infections due to MDR include the potential to deliver higher drug concentrations to the site of infection is very interesting. The authors should shedlight the emerging of MDR bacteria and how these bacteria aquired the resistance in aseperate 2 paragrapghs

Author Response

Thank you for your interesting suggestion. We have made changes in the manuscript. However, we think that this could be the subject of another review article.

Reviewer 2 Report

Dear Authors,

This is a worthy submission filling a certain knowledge gap in the seminal literature.

Its core bottle neck weakness is overt homogeneity of evidence base referign to mostly sources comign from wealthy OeCD economies.

In roder to grasp broader findigns and make them applciable to the wider context of the Global SOuth, LMICs nations and Emerging markets, one has to diversify citations track record and make them more heterogeneous.

Thus I warmly recommend consideration for inclusion of several sources below in the introduction and discussion sections:

https://www.mdpi.com/2079-6382/9/2/57

https://www.sciencedirect.com/science/article/pii/S1473309918303104

I remain willing to review the revised manuscript version.

Author Response

Relevant reference has been added.